# Autogenic Training in Mental Disorders: What Can We Expect?

**DOI:** 10.3390/ijerph20054344

**Published:** 2023-02-28

**Authors:** Dagmar Breznoscakova, Milana Kovanicova, Eva Sedlakova, Maria Pallayova

**Affiliations:** 1Department of Social and Behavioural Medicine, Faculty of Medicine, Pavol Jozef Safarik University, Trieda SNP 1, 040 11 Kosice, Slovakia; 2Center for Mental Functions, Crystal Comfort, LLC, M. R. Stefanika 2427, 093 01 Vranov nad Toplou, Slovakia; 32nd Department of Psychiatry, University Hospital of Louis Pasteur, Rastislavova 43, 041 90 Kosice, Slovakia; 4Department of Pathological Physiology, Faculty of Medicine, Pavol Jozef Safarik University, Trieda SNP 1, 040 11 Kosice, Slovakia; 5Department of Human Physiology, Faculty of Medicine, Pavol Jozef Safarik University, Trieda SNP 1, 040 11 Kosice, Slovakia

**Keywords:** autogenic training, mental disorders, relaxation therapy

## Abstract

Autogenic training (AT) is a well-established self-induced relaxation technique based on autosuggestion. From the past two decades, an increasing number of AT studies strongly suggests the practical usefulness of psychophysiological relaxation in the area of medicine. Despite this interest, to date, limited critical clinical reflection on the application and effects of AT in mental disorders exists. The present paper reviews psychophysiological, psychopathological, and clinical aspects of AT in persons with mental disorders with emphasis on implications for future research and practice. Based on a formal literature search, 29 reported studies (7 meta-analyses/systematic reviews) were identified that examined the effects and impact of AT on mental disorders. The main psychophysiological effects of AT include autonomic cardiorespiratory changes paralleled by central nervous system activity modifications and psychological outputs. Studies demonstrate consistent efficacy of AT in reducing anxiety and medium range positive effects for mild-to-moderate depression. The impact on bipolar disorders, psychotic disorders, and acute stress disorder remains unexplored. As an add-on intervention psychotherapy technique with beneficial outcome on psychophysiological functioning, AT represents a promising avenue towards expanding research findings of brain–body links beyond the current limits of the prevention and clinical management of number of mental disorders.

## 1. Introduction

The burden of mental disorders continues to grow with a profound impact on the wellbeing of people around the world. While effective treatments and social support are available in many countries, access to them is limited. Autogenic training (AT) is a well-established self-relaxation technique based on autosuggestion related to passive concentration of bodily perceptions of heaviness and warmth accompanying a slow breath, which elicits a psychophysiological relaxation response [1]. Autogenic training was developed by the German psychiatrist Dr. Johannes Heinrich Schultz, who published the very first evidence-based book about this method in 1932 [2]. Autogenic training belongs to a broader group of methods of autogenic therapy.

Autogenic therapeutic approaches are methods based on the homeostatic self-regulatory brain mechanisms to support and facilitate existing natural self-healing mechanisms, and thus help to regulate and readjust a variety of functional disorders (e.g., stress or trauma) [3,4]. Autogenic therapy utilizes various methods and their combinations to reduce the disturbing consequences of stimulation. Specifically, the autogenic therapy comprises autogenic training, autogenic neutralization, meditative exercises, autogenic modification, graduated active hypnosis, autogenic feedback training, and autogenic behavior therapy [5]. Some of this self-regulatory neutralization or recuperation occurs naturally during sleep, specifically during rapid eye movement sleep, closely associated with dreaming [6]. We examine these definitions carefully to discover more information about the psychophysiology and the effects of autogenic therapeutic approaches in our findings in the Results section. Figure 1 captures the essence of AT—combinations and procedural interaction with some other methods applied in autogenic therapy [5]. The standard exercises (SE) consist of the passive concentration of the body’s own sensations and attempt to improve self-regulation to optimize homeostatic mechanisms [7]. These exercises must be learned progressively and may be taught in a group or individually. In addition to their clinical applications, regular practice of autogenic methods plays an important role in everyday life (e.g., in the areas of education, industry, sports, and creativity) both to improve the efficiency of various bodily and/or mental activities and as a useful psychophysiological stress buster [5,8].

From the past two decades, an increasing number of AT studies strongly suggests the practical usefulness of psychophysiological relaxation in the area of medicine. The beneficial effects of AT have been observed in many somatic disorders, including but not limited to migraine, chronic pain, essential arterial hypertension, preeclampsia, coronary heart disease, bronchial asthma, unspecified type of somatoform pain disorder, Raynaud’s disease, and functional sleep disorders [1,3,9,10]. Of importance, AT also improved the subjective perception of physical and psychological health and helped to reduce anxiety levels during the COVID-19 crisis [11,12]. Despite this interest, to date, there has been only limited critical clinical reflection on the application and effects of AT in mental disorders.

The present paper reviews psychophysiological, psychopathological, and clinical aspects of AT in persons with mental disorders. This review focuses on advanced understanding of the effects of AT in mental disorders with emphasis on implications for future research and practice. The practical aspects are preceded by a theoretical overview of autonomic dysfunction in mental disorders and psychophysiology of AT. Finally, the review highlights some promising new strategies for future targeted research to advance science, promote efficacious clinical practice, and thereby ultimately optimize care for persons with mental disorders.

## 2. Materials and Methods

Based on the search, appraisal, synthesis, and analysis (SALSA) framework [13], the present article represents a narrative literature review with several components of critical review. References were identified through searches of the PubMed electronic database for articles published in English from inception until December 2022 for relevant studies. A methodical selection of search terms based on PubMed’s Medical Subject Headings (MeSH) was carried out prior to the search to build a search string that generated the best possible result. The main search terms (“Autogenic Training” (Mesh) AND “Mental Disorders” (Mesh)), originally selected by the authors were searched in MeSH to identify additional synonyms, related terms, and subterms. In addition to MeSH, searching was carried out using a combination of keywords searched in the fields “Title/Abstract” to ensure the best possible information retrieval. These searches were supplemented by reviewing and examining the reference lists of all relevant articles included. In addition, the expertise of the review team to identify further studies was employed. Searches were not restricted by study design. To be included in the review, AT procedures must have been performed either in the conventional way (with its 6 SE in the standard combination and sequence: 1—heaviness, 2—warmth, 3—calm and regular heart function, 4—self-regulation of respiration, 5—warmth in the upper abdomen area, and 6—agreeable cooling of the forehead; Figure 1) or using the modified techniques. A more detailed description of AT procedures and the psychophysiology of AT are provided in the Results section. No limitation regarding the number and duration of AT sessions was considered. Similarly, no limitations regarding the length, the place, the manner of the intervention (posture for the exercises), or the adaptation of AT to each pathology were considered.

Based on this formal literature search, 29 reported studies (including 7 meta-analyses/systematic reviews) were identified that examined effects and impact of AT on mental disorders (Table 1).

## 3. Results

### 3.1. Autonomic Dysfunction in Mental Disorders

The etiology of mental disorders is multifaceted and complex. The biopsychosocial model considers the interaction of biological, psychological, and social factors that may be contributing to the development of mental illnesses; there are suggestions for dynamic three-dimensional gene-by-environment-by-time interactions [39]. Underlying risk factors include but are not limited to changes in neuropsychology, neuroanatomy, genetics, and neurochemistry that interplay in the development of mental disorders [39]. As a result, changes in autonomic nervous system (ANS) activity have been observed [40]. The role of ANS is critical to maintaining homeostasis—the stable and balanced internal environment of an organism during both resting activities and physical or emotional stress.

Autonomic dysfunction has been observed in a range of mental disorders, including schizophrenia, bipolar disorder, depressive disorders, anxiety disorders, obsessive–compulsive disorder, posttraumatic stress disorder, and substance dependence [40,41,42,43,44,45,46,47]. Of importance, the association between psychiatric illness and cardiac autonomic dysfunction is largely driven by psychotropic medication use, mainly antidepressant and antipsychotic use [48,49].

Alvares and colleagues [40] meta-analyzed data from 170 studies that examined changes in ANS functioning in subjects with psychiatric disorders compared to controls. The findings strongly suggest that reductions in ANS functioning are associated with psychiatric disorders compared to controls, irrespective of psychotropic medication use; the association was most robust in the subgroup with psychotic disorders [40]. One of the limitations of this study is the use of a single outcome—heart rate variability (HRV)—without the respiratory activity analysis to quantify ANS output. Furthermore, it is well known that sex, age, caffeine, nicotine and alcohol intake, habitual physical activity levels, and body mass index may modify HRV. Therefore, methodological variations may influence estimates of vagal control and thus limit the external validity of the results. Further studies are needed to also examine the relative effect of mental and physical health disorders and their treatment to determine the overall and specific impact of psychiatric illness on HRV. Considerable medicated comorbidity and overlap between many disorders remain to be addressed in future research. In the meantime, cardiovascular preventative approaches need to be implemented as part of the complex management of individuals with mental disorders early in illness progression. The AT can help to target modifiable cardiovascular risk factors to attenuate this risk in individuals with mental disorders [43].

### 3.2. The Psychophysiology of Autogenic Training

The AT self-induced relaxation technique is based on autosuggestion. The autosuggestion is a psychophysiological form of psychotherapy that individuals carry out on themselves using passive concentration and certain combinations of mental and physical stimuli [4,7]. Autogenic training is based on three main psychophysiological principles: (i) reduction of exteroceptive and proprioceptive afferent stimulation; (ii) mental repetition of psychophysiologically adapted verbal formulae; and (iii) ‘passive concentration,’ or complete, effortless mental immersion in the task [7]. Therefore, AT is based on a strict progression of 6 SE as described by Johannes Schultz and his student Wolfgang Luthe [50]. The exercises are usually taught in several sessions and involve passive concentration on breathing, heartbeat and warmth, and heaviness of body parts. Consequently, AT may be used to relax the body by relaxing the mind [4].

The AT procedures may vary, utilizing, e.g., modified AT phrases, tape recorded AT + music = taped AT phrases, synthesized with music (Luthe’s technique), along with 6 SE [21]:
Heaviness, warmth, breathing, personal formulae (modified Schultz technique);Heaviness, warmth, coolness, calmness, and peaceful relaxation (Luthe’s technique);Heaviness and warmth (Schultz and Luthe’s technique);Heaviness (Budzynski’s technique—taped);Heaviness, warmth, breathing patterns, neck and shoulders, mental picture of body alignment (Schultz technique);Heaviness and warmth (Luthe’s technique).

In the United Kingdom, clients are taught six simple mental formulae designed to induce a calm state of mind and body, five additional emotional expression exercises, and individually tailored personal formulae for supporting positive change [18]. Each session can be practiced in a position chosen amongst a set of recommended postures (e.g., lying down, sitting meditation, sitting like a rag doll). The procedure usually takes only about 15 to 22 min and ideally should be practiced daily. Autogenic training is the foundation for all other autogenic therapeutic approaches and is therefore the most important and widely used technique. Even if used alone as the only autogenic therapy, it is sufficient for many treatment situations [5].

In 10–20% of patients, a more intensive method called autogenic neutralization may be necessary; autogenic modification, autogenic feedback training, and autogenic behavior therapy (Figure 1) are newer techniques that are the result of interdisciplinary interaction [5]. The autogenic neutralization might promote corrective, unloading discharges of the psychophysiological disturbing loads that result in neuronal over-excitation [5]. The therapeutic effect is achieved by the neutralization of traumatic emotional experiences/disturbing intrapsychic stimuli through free association and the progressive reorganization of the psychic structure to include previously unacceptable mental contents [5]. Through facilitation of abreactive experiences and verbalization of internal disturbances, such neutralization of the disturbing material is made possible [5]. Both intervention methods require proper training and supervision of the therapeutic process.

The main psychophysiological effects of brain–body interaction relaxation techniques include autonomic cardiorespiratory changes paralleled by central nervous system (CNS) activity modifications and psychological outputs.

#### 3.2.1. Effects of Autogenic Training on Autonomic Nervous System

The psychophysiological changes during AT are closely associated with the induced voluntary control of slow breathing. The respiratory sinus arrhythmia is characterized by an increase in heart rate with inspiration and decrease during expiration [51]. As a result, during deep respiration, the heart rate increases and decreases with each respiratory cycle by as much as 30% [51]. Slow breathing with the prolonged exhalation results in the spillover signals from the medullary respiratory center into the adjacent cardioinhibitory center (depressor area) that cause an increase in the number of impulses transmitted through the vagus nerves to the heart sinus node [51]. The underlying mechanisms also include an increase in the intrathoracic pressure during the prolonged exhalation, which inhibits venous return to the heart resulting in both reduced atrial expansion and reduced baroreceptor activation [51]. As a result, vagal tone is not suppressed during exhalation compared to inhalation so that it can exert its ability in decreasing heart rate. Parasympathetic stimulation hyperpolarizes the membrane potential of the autorhythmic cell and slows depolarization, decreasing the heart rate [51].

#### 3.2.2. Effects of Autogenic Training on Central Nervous System

Electroencephalogram (EEG) studies show a global increase in low-frequency alpha power (a marker of internal attention) and a local decrease in theta power (a marker of an advanced meditative state) [52]. Park and Park [53] demonstrated that personality traits such as harm avoidance, novelty seeking, persistence, self-directedness, and self-transcendence positively correlated with EEG alpha power over the whole cortex during slow paced breathing as compared to spontaneous breathing.

#### 3.2.3. Effects of Autogenic Training on Psychological Outputs

A systematic review [52] demonstrated evidence of increased psychophysiological flexibility linking parasympathetic activity, CNS activities related to emotional control and psychological wellbeing in healthy adults during slow breathing techniques. Psychological outputs related to the slow breathing comprised increased comfort, relaxation, pleasantness, vigor and alertness, and reduced symptoms of arousal, anxiety, depression, anger, and confusion [52]. Positive correlation of personality traits with EEG alpha power highlights the need to consider individual differences such as personality traits when slow breathing is offered for clinical or experimental purposes [53].

### 3.3. Clinical Evidence

#### 3.3.1. Effects of Autogenic Training on Anxiety Disorders

Systematic reviews and meta-analyses have demonstrated consistent and significant efficacy of AT in reducing anxiety [1,15,19,21]. Considering the multifaceted etiology of anxiety disorders, both the neurophysiological system and mental processing need to be addressed. Therefore, multiple relaxation methods constitute essential parts of complex therapeutic interventions for anxiety disorders. In particular, the AT and its combinations with other autogenic methods have a significant potential to translate mind–body links into an individually tailored nondrug therapeutic approach to anxiety disorders.

The AT can Induce a meditative state and elicit the relaxation response diametrically opposed to changes elicited by stress [54]. During AT, proprioceptive feedback from the body directly contributes to implicational meaning. Based on the interacting cognitive subsystems model of affect and cognition [55,56,57], implicational meaning does not map directly onto language as it is linked to emotions paralleled by senses or feelings that arise with implicit meaning content.

According to Wells’ self-regulatory executive function model of generalized anxiety disorder, specific emphasis is placed on metacognitive beliefs and appraisals in the maintenance of worry [58]. Effects on cognition reflected in gained metacognitive, physiological, and mental flexibility in the face of stressors along with a much deeper awareness all contributed to reducing worry and bringing about clearer thinking [18]. Cognitive impairment [20] and an increased level of rigidity [16] decreased the effectiveness of the AT and were disadvantageous for a successful participation in AT. The predictors of the high effectiveness of AT comprised a moderate decrease of the psychological adaptation level together with the ability of the individuals to control their behavior and perseverance in achieving the goal [16].

The intentional anxiety exercise can reduce reported distress associated with anxious thoughts by creating a detached mindfulness with defusion [18]. A detached view of mental events enables the individual to detach from their negative thoughts and see them as thoughts rather than truths [18]. A widened perspective on daily stressors shifted to seeing problems as a challenge rather than as a threat through learning and consistent re-experiencing of a state of AT relaxation over a period of time. This response appeared to contribute to a reduction in anxiety symptoms [18]. Increased confidence and stable personality changes can follow from regular practice of AT [18,59]. Participants also reported a sense of having a lifeline as a result of acquiring AT [18].

People experiencing significant anxiety often suffer from anxiety-related hypervigilance for bodily sensations. Therefore, anxiety treatments aim to help interrupt cyclical and problematic modes of self-focused processing. Of particular importance is to train an ability to let go of heightened body sensitivity, e.g., through a realization that worrying is neither necessary nor likely to be helpful [18].

A recent study has demonstrated that Schultz’s AT and other relaxation techniques were effective in lowering the COVID-19 anxiety levels of young university students [11]. The findings suggest that relaxation techniques help to reduce the levels of anxiety [1,15,17,19,21,22] and can be an alternative to pharmacotherapy in a pandemic crisis, such as the one being experienced worldwide by COVID-19 [11,12].

#### 3.3.2. Effects of Autogenic Training on Mood Disorders

In a meta-analysis of clinical outcome studies, medium range positive effects of AT were found for anxiety disorders, mild-to-moderate depression/dysthymia, and functional sleep disorders [1]. A meta-analysis of 11 studies showed that AT decreased anxiety and depression; calculations observed a reduction effect of anxiety score by 1.37 points (*n* = 85, SMD= −1.37; 95% CI −2.07 to −0.67) and a reduction effect on the depression score by 0.29 point (*n* = 327, SMD= −0.29; 95% CI −0.50 to −0.07) [15].

Relaxation techniques (including the AT) were more effective at reducing self-rated depressive symptoms than waitlist or no or minimal treatment (5 studies, 136 participants) [27]. However, they were not as effective as psychological (mainly cognitive behavior) treatment in reducing self-rated depressive symptoms [27]. Depression symptoms in the AT group improved significantly [26,28] more than in the delayed-treatment control group, but significantly less than in the psychotherapy group [28]. The psychopathological status improved significantly in 23 multimorbid, gerontopsychiatric patients during the 7-week supportive AT course (brief psychiatric rating scale *p* < 0.001; geriatric depression scale *p* < 0.001) [29]. Bowden and colleagues [17] have demonstrated that AT significantly reduced anxiety and depression and improved sleep patterns for patients with various health conditions. These findings suggest that AT may help tackle the vicious triangle of insomnia, anxiety, and depression, and this novel approach could be incorporated into primary care [17].

A recent study examined the attention efficiency as determined by the perceptual speed index measured using the attention and perceptiveness test in people with depression [24]. The results showed that a 15-minute-long, one-time AT session improved the efficiency of attention and perceptiveness in individuals diagnosed with depressive disorders [24]. The pilot study by Ramirez-Garcia and colleagues [25] has demonstrated that participants affected by human immunodeficiency virus (HIV) reported better emotion management and improvements in depressive symptoms and quality of life following the AT intervention. The findings suggest that a randomized trial to test the effectiveness of progressive muscle relaxation and AT on depressive symptoms and quality of life in people living with HIV is feasible [25].

Finally, emotional disorders are common among adolescents. A recent study [23] has demonstrated that both two 8-week mind-body interventions, namely mindfulness-oriented meditation training and AT, resulted in an increased level of adolescents’ cooperativeness (i.e., the acceptance of others, compassion, forgiveness) and reduced emotional difficulties (e.g., feeling unhappy, downhearted). The increases in cooperativeness following the AT correlated well with reductions in negative emotional symptoms [23].

To date, to our knowledge, there is no evidence published examining effects of AT on bipolar disorders.

#### 3.3.3. Effects of Autogenic Training on Schizophrenia and Psychotic Disorders

Shibata and Motoda [32] demonstrated that the application of AT to subjects with schizophrenia produced favorable effects on psychological tests results and clinical symptoms. In further research, their results suggested through visual evoked responses recordings that the brain activity of patients with schizophrenia may be stabilized by practicing AT [30]. While the AT has been regarded as one way of rehabilitating schizophrenic patients by making them willing to adjust themselves to reality and assisting their regaining their confidence through enhancing self-esteem, limits for the application of AT to schizophrenia exist [31]. Currently, there is only one review published on effects of mind–body medicine on psychotic disorders [60]. Although supportive evidence was found for music therapy, meditation, and mindfulness techniques, no clinical trials were found for AT in this population [60]. Further research is necessary to assess the validity of applying AT in the treatment of schizophrenia and psychotic disorders.

#### 3.3.4. Effects of Autogenic Training on Trauma- and Stressor-Related Disorders

Both posttraumatic stress disorder (PTSD) and acute stress disorder are marked by increased stress and anxiety following exposure to a traumatic or stressful event. Autogenic training significantly decreased cardiac sympathetic nervous activity and significantly increased cardiac parasympathetic nervous activity along with a significant decrease in the total points of impact of event scale—revised questionnaire in fire service workers with PTSD [35]. In a randomized controlled trial, the intervention group of postwar Kosovar adolescents with PTSD undergoing a mind–body skills group program including AT showed a significant decrease in PTSD symptom scores (measured using the Harvard trauma questionnaire) following intervention that was maintained at 3-month follow-up [34]. To date, there is no published evidence available on the effects of AT on acute stress disorder.

Jojic and Leposavic demonstrated a significant decrease in values of arterial blood pressure, pulse rate, concentration of cholesterol and cortisol following AT intervention as the sole therapy in adolescents [36] and in middle-aged adults [37] diagnosed with adjustment disorder.

A systematic review of the literature was carried out to examine the effect of stress management techniques, specifically of progressive muscle relaxation (PMR), AT, and guided imagery (GI), on stress levels of individuals with addictive behaviors [33]. The findings suggest that PMR might lead to a reduction of stress levels in persons with addictive behaviors, while no such evidence was found concerning GI and AT [33].

#### 3.3.5. Effects of Autogenic Training on Other Mental Disorders

Insomnia, dyssomnia, and other sleep-related problems are commonly associated with depressive disorders, anxiety disorders, or psychological distress. Several studies have consistently demonstrated that AT may improve sleep patterns and increase sleep quality in persons with various health conditions, including functional sleep disorders [1,3,17]. The findings implicate that AT as a widely available self-relaxation procedure could be incorporated into primary care to improve insomnia in many adults, and thus deprescribe sleep medications.

Sexual dysfunction is a common problem among both men and women due to many physical and/or psychological causes. Stanton and colleagues [38] investigated the impact of AT on premenopausal women who self-reported decreased or absent sexual arousal for at least 6 months. The results indicated that one brief (22-min) session of AT significantly improved acute subjective arousal and increased perceived genital sensations from the pre-manipulation erotic film to the post-manipulation erotic film in fertile premenopausal women with self-reported arousal concerns [38].

## 4. Discussion

### 4.1. Clinical Gaps

Anxiety-, mood-, and stress-disorders are important challenges worldwide. Identifying novel treatments for mental disorders is urgently needed and is important for tackling their cardiovascular, metabolic, and other somatic sequelae. Given the dramatically increased number of persons affected by mental disorders, it is becoming even more critical to countries worldwide and their future to successfully address challenges associated with mental illnesses and their complications. While the present findings (Table 1) have important implications, particularly for treatment of anxiety and depression symptoms, more research evidence is needed with emphasis on separate mental disorders over longer periods of follow-up. To date, there is still no published evidence on the effects of AT on bipolar disorders, psychotic disorders, and acute stress disorder. Future research should also address the development of randomized controlled trials assessing the effects of AT on different dimensions of anxiety and depression.

The adaptation of AT to each pathology, whether it is carried out in a guided or autonomous, virtual or face-to-face manner, are aspects that may be relevant for the conclusions. After careful reading reports of the reviewed studies, the complete relevant information regarding AT was available in 7 out of 29 papers. Majority of the studies exploring the effects of AT were pragmatic—designed to evaluate the effectiveness of AT in real-life routine practice conditions rather than under optimal situations. Although the detailed description of AT procedures, including the manner of AT is often missing in the published papers, pragmatic studies usually produce results that can be generalized and applied in routine practice settings.

Recent results of the effects of AT on autonomic nervous system imbalance in mental disorders have been very promising. Yet, methodological variations may influence estimates of sympathovagal control and thus limit the external validity of the results. To examine the overall and specific impact of mental and physical health disorders and their treatment on autonomic function, more accurate autonomic function testing is needed. Specifically, for ANS monitoring, HRV should include respiratory parameters. Future studies should examine resting and dynamic ANS balance and ANS imbalances even in asymptomatic patients with mental disorders in order to individually tailor protective cardiovascular treatments early in illness progression.

Previous studies have demonstrated an independent bidirectional link between depression and cardiovascular comorbidity [61,62]. Of particular importance is an independent twofold to threefold increased risk of future cardiac events in individuals with coronary artery disease and depression compared to those without depression [61]. According to the results of the large case controlled INTERHEART study, the presence of psychosocial stressors was the third most important risk factor (odds ratio of 2.67) for developing cardiovascular disease [63]. Following cardiac events, individuals with mental illness experience a 14% lower rate of invasive coronary interventions (47% in the case of schizophrenia) and they have an 11% increased mortality rate [64]. A strong co-occurrence of mood disorders and coronary artery disease is accompanied by a reciprocal worsening of the prognosis for the two conditions [65]. Autogenic training might be a part of the solution, as it has been reported to improve psychological wellbeing in chronic medical patients [14].

Currently, the practice of AT is also recommended to people who live in fear or in anxiety, and to those who are afraid of illness or need to improve the quality of relationships with others [12].

### 4.2. Future Directions

The majority of participants values AT highly and considers it helpful in maintaining psychological health [12]. Therefore, achieving emotional stability is one of the main reasons for application of AT in clinical psychology [4].

Future research should aim at using the AT as an intervention to also improve autonomic function, and thus control (prevent, slow down, delay, or reverse) the progression of cardiometabolic complications. If this approach were successful, it would translate into fewer subjects requiring somatic treatments and fewer people progressing to cardiometabolic complications, which is an enormous benefit to individuals and society.

Future studies should also be designed and conducted to test the hypothesis that reduced vagal tone may be an endophenotype for the development of psychotic symptoms in individuals with schizophrenia [66] and a marker for future cardiovascular and other stress-related disease [67]. Dynamic changes in HRV might represent a useful phenotype for psychological and physical comorbidities [68]. If proved in future studies, its testing would be advised to help detect causes and assess the efficacy of prevention and intervention therapies in several psychosomatic and psychiatric dysfunctions on a timely basis. The assessment of the durability and long-term effects of AT treatment on cardiovascular health status will also facilitate development of more efficient, safer, and more cost-effective treatment methods and diagnostic tools.

## 5. Conclusions

Autogenic training is a widely available self-relaxation procedure with beneficial outcomes on psychophysiological functioning. A strong co-occurrence of the mental illness and cardiometabolic complications highlights the need to diagnose and treat these frequent comorbid conditions early. Autogenic training as an add-on intervention psychotherapy technique represents a promising avenue towards expanding and translating future research findings of brain–body links beyond the current limits of the prevention and clinical management of number of mental disorders. Further clinical research studies in well-characterized patient populations are needed to improve and guide our current understanding of the pathophysiology of mental disorders and to allow identification of novel potential biomarkers and prognostic factors of disease progression based on responses to autogenic therapy interventions. This will also be essential for the identification of patients who may benefit best from an intervention. It is expected that the knowledge to be gained from future studies will inform interventions for mental disorders treatment and help ensure incorporation of findings into psychiatric care pathways and provision of funding to take the intervention directly into clinical practice.

## Figures and Tables

**Figure 1 ijerph-20-04344-f001:**
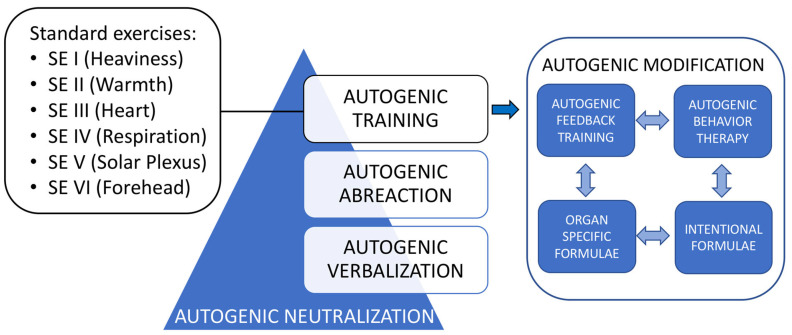
Autogenic training: combinations and procedural interaction. Adapted from Luthe 1979 [5]. SE: standard exercises.

**Table 1 ijerph-20-04344-t001:** Reported studies concerning effects of autogenic training on mental disorders (categorized and sorted chronologically, from most recent to least recent).

Reported Study	Number of Studies	Number of Participants	Outcomes	Results
Anxiety disorders				
de Rivera et al., 2021 [12]	1	75	The effectiveness of autogenic training (AT)	There was an increase in the practice of AT during the COVID-19 pandemic. Autogenic training was very useful for physical and psychological health and for a better understanding of others.
Ramirez-Garcia et al., 2020 [14]	A protocol for a systematic review	Randomized controlled trials to be identified	The efficacy of AT on psychological wellbeing (anxiety, distress, depression) in people with chronic physical health problems	Pending results.
Ozamiz-Etxebarria et al., 2020 [11]	1	44	Generalized anxiety disorder-7 (GAD-7) scale	Jacobson’s progressive relaxation techniques, Schultz’s AT, abdominal relaxations, and visualizations were effective in lowering the COVID-19 anxiety levels of university students as an alternative to pharmacotherapy.
Seo and Kim, 2019 [15]	Systematic review: 21 meta-analysis: 11	85–327	Anxiety score	Autogenic training decreased anxiety score.
Aivazyan and Zaitsev, 2018 [16]	1	325	The effectiveness of AT	Autogenic training reduced anxiety, tension, negative feelings, stress sensitivity, and improved mood and activity levels in patients affected by chronic somatic diseases.
Bowden et al., 2012 [17]	1	153	“Measure Yourself Medical Outcome Profile” and hospital anxiety and depression scale	Autogenic training reduced anxiety.
Yurdakul et al., 2009 [18]	1	12	Autogenic training experience in anxiety	The cognitive changes reported have implications for anxiety treatments.
Manzoni et al., 2008 [19]	27	1014	Anxiety, psychometric questionnaires	There was consistent and significant efficacy of relaxation training in reducing anxiety.
Kircher et al., 2002 [20]	1	16	Effectiveness of the AT in cognitively impaired, frail elderly	Mentally impaired, frail elderly participants were able to learn the AT. Cognitive impairment was disadvantageous for successful participation.
Stetter and Kupper, 2002 [1]	60	>1500	Anxiety	Medium range positive effects of AT were found for anxiety disorders.
Ernst and Kanji, 2000 [21]	8	245	Anxiety, Spielberger’s state–Trait anxiety inventory	Autogenic training reduced stress and anxiety.
Sakai, 1997 [22]	1	55	The impact of AT on anxiety disorders	Twenty-eight patients with anxiety disorders (51%) were cured, fourteen (25%) much improved, eight (15%) improved, and five (9%) unchanged at the end of the treatment. Forty-two patients (76%) were assessed as having had successful treatment.
Mood disorders				
Feruglio et al., 2022 [23]	1	72	Cooperativeness and emotional symptoms (self-reported temperament and character inventory; strengths and difficulties questionnaire for adolescents)	Both mindfulness-oriented meditation training and AT enhanced a cooperative attitude in adolescents and helped reduce their emotional problems.
Rucka and Talarowska, 2022 [24]	1	42	Attention efficiency determined by the perceptual speed index measured using the attention and perceptiveness test	A 15-min-long, one-time AT session improved the efficiency of attention and perceptiveness in people with depression.
Ramirez-Garcia et al., 2020 [25]	1	42	Emotion management, depressive symptoms, and quality of life	Adults living with human immunodeficiency virus reported better emotion management and improvements in depressive symptoms and quality of life following the AT intervention.
Seo and Kim, 2019 [15]	systematic review: 21 meta-analysis: 11	85–327	Depression score	Autogenic training decreased depression score.
Bowden et al., 2012 [17]	1	153	“Measure Your Medical Outcome Profile” and hospital anxiety and depression scale	Autogenic training reduced depression.
Goto et al., 2009 [26]	1	1	Self-rating depression scale, manifested anxiety scale	Depressive state was markedly alleviated following the AT.
Jorm et al., 2008 [27]	systematic review: 15 meta-analysis: 11	48–286	Depressive symptoms	Relaxation techniques (including the AT) were more effective at reducing self-rated depressive symptoms than no or minimal treatment.
Morgan and Jorm, 2008 [28]	2	189	Depressive symptoms	Depression symptoms in the AT group improved significantly more than in the control group, but significantly less than in the psychotherapy group.
Stetter and Kupper, 2002 [1]	60	>1500	Depression/dysthymia	Medium range positive effects of AT were found for mild-to-moderate depression/dysthymia.
Kircher et al., 1997 [29]	1	23	Psychopathological status: brief psychiatric rating scale, geriatric depression scale, the cognitive state (a mini-mental state examination)	Autogenic training was a useful component in psychotherapeutic and psychiatric therapy for elderly multimorbid in- and outpatients.
Schizophrenia and psychotic disorders				
Motoda et al., 1969 [30]	1	30	Visually evoked responses	Recordings of electrical signals generated by the visual cortex in response to visual stimulation suggest the brain activity in schizophrenia may be stabilized by practicing AT.
Shibata, 1968 [31]	1	1	Limits of AT application to schizophrenia	The limits of AT application depend on the selection of schizophrenic patients. The patients in remission should be selected for AT.
Shibata and Motoda, 1967 [32]	1	65	Rehabilitation effects of AT on schizophrenic convalescent patients	All study participants progressed favorably during the SE. After proceeding on to meditation exercises, symptoms aggravated in several patients.
Trauma- and stressor-related disorders				
Louvardi et al., 2021 [33]	4	188	The effect of stress management on stress levels measured using instruments or biochemical assessments	Progressive muscle relaxation might lead to a reduction of stress levels in persons with addictive behaviors, while no such evidence was found concerning guided imagery and AT.
Gordon et al., 2008 [34]	1	82	Posttraumatic stress disorder (PTSD) symptom scores (Harvard trauma questionnaire)	Postwar Kosovar adolescents with PTSD undergoing AT experienced a significant decrease in PTSD symptom scores that was maintained at 3-month follow-up.
Mitani et al., 2006 [35]	1	22	PTSD signs and symptoms (cardiac sympathetic and parasympathetic nervous activity, impact of event scale—revised questionnaire)	Autogenic training ameliorated disturbances in cardiac autonomic nervous activity and improved self-reported psychological dysfunction secondary to PTSD (intrusion, avoidance, hyperarousal) in fire service workers with PTSD.
Jojic and Leposavic, 2005 [36]	1	31	Adjustment disorder indicators (arterial blood pressure, pulse rate, concentration of cholesterol and cortisol)	Autogenic training significantly decreased the values of physiological indicators of adjustment disorder and diminished the effects of stress in adolescents with adjustment disorder.
Jojic and Leposavic, 2005 [37]	1	35	Adjustment disorder indicators (arterial blood pressure, pulse rate, concentration of cholesterol and cortisol)	Autogenic training significantly decreased the values of physiological indicators of adjustment disorder and diminished the effects of stress in adults with adjustment disorder.
Other mental disorders				
Litwic-Kaminska et al., 2022 [3]	1	22	Sleep quality (the Pittsburg sleep quality index), physiological stress reactions	Sleep quality significantly increased after two-week AT usage in experimental group.
Stanton et al., 2018 [38]	1	25	Genital sexual arousal, subjective sexual arousal, and perceived genital sensations	Autogenic training significantly improved acute subjective arousal and increased perceived genital sensations in premenopausal women with self-reported arousal concerns.
Bowden et al., 2012 [17]	1	153	Sleep questionnaires	Autogenic training improved sleep patterns.
Stetter and Kupper, 2002 [1]	60	>1500	Functional sleep disorders	Medium range positive effects of AT were found for functional sleep disorders.

## Data Availability

No new data were created or analyzed in this study. Data sharing is not applicable to this article.

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
