# Peer review of "Autogenic Training in Mental Disorders: What Can We Expect?"

_ijerph, 2023, doi:10.3390/ijerph20054344_

Round 1

Reviewer 1 Report

From the results of previous studies described in Table 1, it would be good to present quantitative results together.

If possible, it would be better to present Table 1 in chronological order by dividing it into autonomic dysfunction in mental disorders and psychophysiology of autogenic training, as presented in the results of the main text.

Author Response

Response to Reviewer 1 Comments

English language and style: Moderate English changes required

Response: Two native English-speakers (Prof. Alan R. Schwartz, MD and Roy M. Vigneuelle, PhD – both highly experienced scientists based in the US) have checked the English. The manuscript has been revised accordingly.

Point 1: From the results of previous studies described in Table 1, it would be good to present quantitative results together.

Response 1: We thank the Reviewer for this important remark. To present quantitative results (of previous primary studies) together, a systematic review and meta-analysis would have to be conducted. In order to calculate a summary statistic for each study to describe the observed intervention effect for every study, raw data would have to be obtained first. Only afterwards, a summary (combined) intervention effect estimate can be calculated as a weighted average of the intervention effects estimated in the individual studies. This process, if feasible, usually takes a comprehensive effort over weeks or even months. Since it was not feasible to perform a systematic review because of the restriction of time and other resources, we prepared a narrative review that can still be of great value to the clinical and scientific community. It can also be a helpful precursor to future systematic reviews (and meta-analyses) where critical appraisal of each included study is required.

The purpose of the present paper was to provide a comprehensive and comparative overview (with critical clinical reflection) of the application and psychophysiological effects of AT in mental disorders. The article represents a narrative literature review with several components of critical review and umbrella review to give a high-level overview. We believe the article can be useful in determining how to best translate the evidence into practice. Also, reading the article may help to answer multiple questions of the readers - what is known on a topic, what remains unknown, how the evidence is interpreted to guide practice, and recommendations are made for what requires further research.

Point 2: If possible, it would be better to present Table 1 in chronological order by dividing it into autonomic dysfunction in mental disorders and psychophysiology of autogenic training, as presented in the results of the main text.

Response 2: We appreciate the Reviewer suggestion. Actually, the Table 1 was originally designed to only include studies reporting results of the effects of AT on mental disorders. It is to complement the Results section 3.3. Clinical evidence. The parts 3.1. Autonomic dysfunction in mental disorders and 3.2. The psychophysiology of autogenic training contain only a few core studies described in the main text. We believe these data would be more visually appealing if presented within the text.

In order to present Table 1 more effectively (making the table more attractive and the data more easy to understand), we have slightly revised the table. Specifically, we split up the table that was too cluttered and too long. Five subsections were created to group studies chronologically by categories (Anxiety disorders, Mood disorders, Schizophrenia and psychotic disorders, Trauma- and stressor-related disorders, and Other mental disorders). We have added 3 studies (the effects of AT in schizophrenia) into the revised table. In the original Manuscript, they were mentioned only in the main text. We have removed the study by Schilling et al. (2018) from the main text and from the table, as it utilized only the progressive muscle relaxation therapy modality without AT. Therefore, the final number of the included studies changed from 27 to 29 (Lines 21, 105, 370). If the Editor/Referee preferred another way of presentation of the table, we would undertake further revisions if required.

To avoid data/text repetition (written in the main text, verbatim not to be repeated in the table), the text has been revised accordingly.

Reviewer 2 Report

This paper deals with an interesting and highly topical subject. However, in my opinion, there are some aspects of the methodology that should be improved. 

1.- Mental disorders represent a very broad group of pathologies with very different symptoms and characteristics. Although reference is made to some of them, I consider that the conclusions cannot be extrapolated to all mental disorders. It would be necessary to narrow down the subject in order to be able to provide relevant data in the introduction and to be able to classify the results. 

2.- A systematic review of the literature, or a review of systematic reviews, where appropriate, is recommended. The methodology used only collects publications from a database and does not take into account their methodological quality, which greatly limits the validity of what is analysed. 

3.- The adaptation of AT to each pathology, whether it is carried out in a guided or autonomous, virtual or face-to-face manner, are aspects that should be specified in the description of the publications included, as they may be relevant for the conclusions.

Author Response

Response to Reviewer 2 Comments

English language and style: Extensive editing of English language and style required

Response: We regret there were problems with the English. The paper has been carefully revised by two native English-speakers to improve the grammar and readability (Prof. Alan R. Schwartz, MD and Roy M. Vigneuelle, PhD – both highly experienced scientists based in the US). Original Word documents with tracked changes made by the above-mentioned native English-speakers can be forwarded to the Editorial Office on request.

Prof. Alan R. Schwartz, MD was my mentor at Johns Hopkins University, Baltimore, Maryland, USA (2008-2010). Currently, he serves as Adjunct Professor, University of Pennsylvania Perelman School of Medicine; Part-time Faculty, Vanderbilt University School of Medicine; Professor (ret.), Johns Hopkins University; Distinguished Visiting Professor, Universidad Peruana Cayetano Heredia (Lima, Peru); Scientific Advisor, Sleep and Breathing technologies; and Practitioner and Clinical Investigator, SJMC Medical Group at University Maryland.

Roy M. Vigneuelle, PhD is a former Senior Scientist (now retired). He worked as a Contractor for Military Operational Medicine Research Program, General Dynamics Information Technology, US Army Medical Research and Materiel Command, Fort Detrick, Maryland.

Us, authors, also carefully checked the English and addressed this issue during revision.  

Point 1: Mental disorders represent a very broad group of pathologies with very different symptoms and characteristics. Although reference is made to some of them, I consider that the conclusions cannot be extrapolated to all mental disorders. It would be necessary to narrow down the subject in order to be able to provide relevant data in the introduction and to be able to classify the results. 

Response 1: We thank the Reviewer for this valuable feedback. Our conclusions were carefully written and revised. They are not to be extrapolated to all mental disorders. The conclusions offer a brief summary of the main ideas of the article and convey the larger implications of our study. We believe, the conclusions also offer creative approaches for framing/contextualizing the research problem based on the results of our review. While our article does not offer conclusive evidence applicable to all mental disorders, it does provide answers to the question stated in the title of the article.

The reviewer suggests narrowing down the subject to yield solid conclusions. To date, 7 meta-analyses/systematic reviews were published that examined the effects and the impact of AT on anxiety/anxiety score/anxiety disorders, depression score/depressive symptoms/depression/dysthymia, stress, and functional sleep disorders. We have identified additional 22 primary research studies reporting the effects of AT on mental disorders. Nevertheless, to date, there has been only limited critical research- and clinical psychophysiological reflection that would comprehensively characterize and evaluate the effects of AT in mental disorders. Therefore, we aimed to address this topic in the present review article.

Point 2: A systematic review of the literature, or a review of systematic reviews, where appropriate, is recommended. The methodology used only collects publications from a database and does not take into account their methodological quality, which greatly limits the validity of what is analysed. 

Response 2: We thank the Reviewer for the important insights and remarks. Based on the SALSA Framework, the present article represents a narrative literature review with several components of critical review and umbrella review (suggested by the Reviewer) to give a high-level overview. Systematic search (with a methodical selection of search terms) was implemented to produce the most appropriate evidence synthesis to address broader questions related to the selected topic. Since it was not feasible to perform a systematic review because of the restriction of time and other resources, we prepared the above-mentioned type of review that can still be of great value to the clinical and scientific community. It can also be a helpful precursor to future systematic reviews (and meta-analyses) where critical appraisal of each included study is required. Despite different study designs and methodological quality of the reviewed studies, we believe the article can be useful in determining how to best translate the evidence into practice. Finally, it answers multiple essential questions - what is known on a topic, what remains unknown, how the evidence is interpreted to guide practice, and recommendations are made for what requires further research.

Point 3: The adaptation of AT to each pathology, whether it is carried out in a guided or autonomous, virtual or face-to-face manner, are aspects that should be specified in the description of the publications included, as they may be relevant for the conclusions.

Response 3: We thank the Reviewer for this important point. We agree that the above-mentioned aspects of AT should be specified in the description of the reviewed publications, wherever possible, as they may be relevant for the conclusions. After careful reading reports of the reviewed studies, we found the complete relevant information regarding the aspects of AT mentioned by the reviewer in 7 out of 29 papers. Majority of the studies exploring the effects of AT were pragmatic. While pragmatic studies are designed to evaluate the effectiveness of AT in real-life routine practice conditions, explanatory trials aim to test whether AT works under optimal situations. Although the detailed description of AT procedures, including the manner of AT are often missing in the published papers, pragmatic studies usually produce results that can be generalized and applied in routine practice settings.

In light of the above, we have expanded the Methodology section to further clarify the selection process as follows: “To be included in the review, AT procedures must have been performed either in the conventional way (with its 6 SE in the standard combination and sequence: 1—heaviness, 2—warmth, 3—calm and regular heart function, 4—self-regulation of respiration, 5—warmth in the upper abdomen area, and 6—agreeable cooling of the forehead; Figure 1) or using the modified techniques. A more detailed description of AT procedures and the psychophysiology of AT are provided in the Results section. No limitation regarding the number and duration of AT sessions was considered. Similarly, no limitations regarding the length, the place, the manner of the intervention (posture for the exercises), or the adaptation of AT to each pathology were considered.” (Lines 97-104)

Also, we have revised the Discussion (Clinical gaps) incorporating Reviewer’s remarks as follows: “The adaptation of AT to each pathology, whether it is carried out in a guided or autonomous, virtual or face-to-face manner, are aspects that may be relevant for the conclusions. After careful reading reports of the reviewed studies, the complete relevant information regarding AT was available in 7 out of 29 papers. Majority of the studies exploring the effects of AT were pragmatic - designed to evaluate the effectiveness of AT in real-life routine practice conditions rather than under optimal situations. Although the detailed description of AT procedures, including the manner of AT is often missing in the published papers, pragmatic studies usually produce results that can be generalized and applied in routine practice settings.” (Lines 367-374)

Reviewer 3 Report

Thank you for the opportunity to review this work.  With a few changes I believe it is suitable for publication and adds to the research and knowledge on this topic.

Pg 1 42-48. My concern as one unfamiliar to AT is that you have provided terms but no descriptions as to what these are.  Yes, you delve into that farther down, but without some information this section is weak.  In the least note to your reader that you will delve more into these definitions in your findings.

Pg 2 line 60-  SE Standard exercises- is this a heading? 

P 10 line 331- I appreciate your commentary that the wide variety of AT methods can limit result validity.

Line 353 The majority…

Line 356 Future research…

Author Response

Response to Reviewer 3 Comments

English language and style: English language and style are fine/minor spell check required

Response: We thank the Reviewer and appreciate the feedback. Since the extensive editing of English language and style was required by one of the other Reviewers, two native English-speakers (Prof. Alan R. Schwartz, MD and Roy M. Vigneuelle, PhD – both highly experienced scientists based in the US) have checked the English. The manuscript has been revised accordingly.

Point 1: Pg 1 42-48. My concern as one unfamiliar to AT is that you have provided terms but no descriptions as to what these are. Yes, you delve into that farther down, but without some information this section is weak. In the least note to your reader that you will delve more into these definitions in your findings.

Response 1: Many thanks to the Reviewer for this very important point. We completely agree with the Reviewer. We have revised the text accordingly. Specifically, we have added the following statement: “We will examine these definitions carefully to discover more information about the psychophysiology and the effects of autogenic therapeutic approaches in our findings in the Results section.” (Lines 51-53)

Point 2: Pg 2 line 60-  SE Standard exercises- is this a heading? 

Response 2: We thank the Reviewer for this point. The „SE, the standard exercises“ (Line 60 in the Original Manuscript) is not a heading. It belongs to Figure 1. In figures, all abbreviations should be listed at the end with their definitions. To avoid confusion, we have transferred this explanation of the SE abbreviation below the figure (above the figure caption/title). (Line 63)

Point 3: P 10 line 331- I appreciate your commentary that the wide variety of AT methods can limit result validity.

Response 3: We thank the Reviewer for the positive and encouraging feedback. It is much appreciated.

Point 4: Line 353 The majority…

Response 4: The text has been corrected.

Point 5: Line 356 Future research…

Response 5: The text has been corrected.

Thanks again to the Editorial Office, Editors & Reviewers for the consideration!

Respectfully,

Prof. Maria Pallayova, the corresponding author

In Kosice, Slovak Republic, February 18, 2023.